# Increased Circulating Epithelial Tumor Cells (CETC/CTC) over the Course of Adjuvant Radiotherapy Is a Predictor of Less Favorable Outcome in Patients with Early-Stage Breast Cancer

Matthias Mäurer [1,2,*], Dorothea Schott [3], Monika Pizon [3], Sonia Drozdz [1], Thomas Wendt [1], Andrea Wittig [1] and Katharina Pachmann [3]

[1] Department of Radiotherapy and Radiation Oncology, University Hospital Jena, Bachstraße 18, 07743 Jena, Germany
[2] Clinician Scientist Program OrganAge, Interdisciplinary Center for Clinical Research (IZKF), Jena University Hospital, 07747 Jena, Germany
[3] Transfusionsmedizinisches Zentrum Bayreuth, Kurpromenade 2, 95448 Bayreuth, Germany
[*] Correspondence: matthias.maeurer@med.uni-jena.de

**Abstract:** Background: Adjuvant radiotherapy (RT) is an integral component of a multidisciplinary treatment strategy for early-stage breast cancer. It significantly reduces the incidence of loco-regional recurrence but also of distant events. Distant events are due to tumor cells disseminated from the primary tumor into lymphatic fluid or blood, circulating epithelial tumor cells (CETC/CTC), which can reach distant tissues and regrow into metastases. The purpose of this study is to determine changes in the number of CETC/CTC in the course of adjuvant RT, and to evaluate whether they are correlated to local recurrence and distant metastases in breast cancer patients. Methods: Blood from 165 patients irradiated between 2002 and 2012 was analyzed 0–6 weeks prior to and 0–6 weeks after RT using the maintrac® method, and patients were followed over a median period of 8.97 (1.16–19.09) years. Results: Patients with an increase in CETC/CTC numbers over the course of adjuvant RT had a significantly worse disease-free survival ($p = 0.004$) than patients with stable or decreasing CETC/CTC numbers. CETC/CTC behavior was the most important factor in predicting subsequent relapse-free survival. In particular, patients who had received neoadjuvant chemotherapy were disproportionately more likely to develop metastases when cell counts increased over the course of RT ($p = 0.003$; hazard ratio 4.886). Conclusions: Using the maintrac® method, CETC/CTC were detected in almost all breast cancer patients after surgery. The increase in CETC/CTC numbers over the course of RT represents a potential predictive biomarker to judge relative risk/benefit in patients with early breast cancer. The results of this study highlight the need for prospective clinical trials on CETC/CTC status as a predictive criterion and for individualization of treatment. Clinical Trial registration: The trial is registered (2 May 2019) at trials.gov under NCT03935802.

**Keywords:** biomarkers; circulating epithelial tumor cells; early-stage breast cancer; prediction; radiotherapy

## 1. Introduction

Radiotherapy (RT) is an integral component in the multidisciplinary management of invasive breast cancer [1]. In the neoadjuvant situation, where chemotherapy is applied prior to surgery to reduce the tumor size, RT is used subsequent to surgery. If chemotherapy is given as adjuvant therapy, RT is applied after chemotherapy and may be administered together with hormone therapy, if indicated. However, the impact of adjuvant RT on patient outcomes remains a matter of debate. Thirty years after the widespread use of adjuvant RT, RT unequivocally reduces the risk of loco-regional recurrence. However, this has not been translated into a comparably powerful reduction of long-term mortality, which depends on risk constellation [2]. The aim of breast conserving surgery (BCS) is to remove all detectable

macroscopic disease, but it is possible that some tumor foci may remain in loco-regional tissues. It is assumed that these foci, if untreated, may progress into local, or loco-regional, tumor recurrence. Thus, adjuvant RT is often used following BCS [3,4] and in selected mastectomy cases [5].

Numerous randomized trials and meta-analyses have proven, with the highest level of evidence, the effectiveness of postoperative radiation in reducing the risk of local recurrence [4]. For example, among women included in the SEER cancer registries between 1992 and 2005 [6], those who received both BCS and RT had lower mortality from breast cancer than those not receiving RT. However, although reduction in local recurrences is assumed to prevent distant seeding and reseeding from persistent reservoirs of loco-regional disease [7], distant metastases can occur without prior local recurrence. Thus, it seems that an outcome of RT is also impacted by breast cancer biology [8], but little is currently known as to whether, and how, local RT prevents distant metastases [9].

A prerequisite for distant metastases are cells that leave the primary tumor and circulate in peripheral blood [10,11]. If distant metastases develop, in spite of RT preventing local relapse, the question arises whether these metastases are due to cells which left the primary tumor at an earlier point in time, or whether local RT can mobilize radioresistant cells into circulation [12].

We, therefore, investigated how circulating tumor cells behave in primary breast cancer after complete resection of the tumor in the course of adjuvant RT and determined whether there is an association between this behavior and risk of local recurrence or metastasis [13]. In contrast to the plethora of publications focusing on circulating tumor cells in metastatic disease, we report, herein, for the first time, on the association between circulating tumor cell behavior over the course of adjuvant RT in early breast cancer and the further course of disease [14–16].

## 2. Materials and Methods

For CETC/CTC enumeration and further characterization, the maintrac® approach was used as previously reported [14]. In brief, 1 ml of blood was subjected to red blood cell lysis using 15 mL of erythrocyte lysis solution (Qiagen, Hilden, Germany) for 15 min at 4 °C. Following this, intact cells were collected by centrifugation at $700\times g$ and re-suspended in 500 µL of PBS-EDTA. Subsequently, 20 µL of this cell suspension was incubated with 20 µL of a mastermix containing 120 µL EpCAM-FITC (fluoroisothiocyanate (FITC)-conjugated anti-EpCAM antibody (CD-326, Miltenyi, Bergisch Gladbach, Germany)), 100 µL 10% BSA, 4 µL 7-AAD (7-Aminoactinomycin D Sigma-Aldrich Co., Taufkirchen, Germany), and 776 µL of PBS-EDTA. The corresponding isotypic control antibody for anti-EpCAM was FITC-conjugated mouse IgG1 (Miltenyi Biotec GmbH), which was used at the same final concentration. Samples were subsequently diluted with 430 µL PBS-EDTA and a defined volume of the cell suspension was transferred to flat-bottom wells of ELISA plates (Greiner Bio-One, Monroe, NC, USA).

The analysis of cellular red and green fluorescence was performed using a fluorescence scanning microscope, ScanR, (Olympus, Tokyo, Japan), enabling the detection and relocation of cells for the visual examination of EpCAM-positivity. For data analysis, we used the ScanR Analysis software (Olympus). Viable CETC/CTC were defined as EpCAM-positive cells, lacking nuclear 7-AAD staining and with intact morphology, and only these cells were counted (Figure 1). Fluorospheres (Flow-Check 770, Beckman Coulter, Brea, CA, USA) were used for the daily verification of optical components and detectors of the microscope, which are required to ensure the consistent analysis of samples.

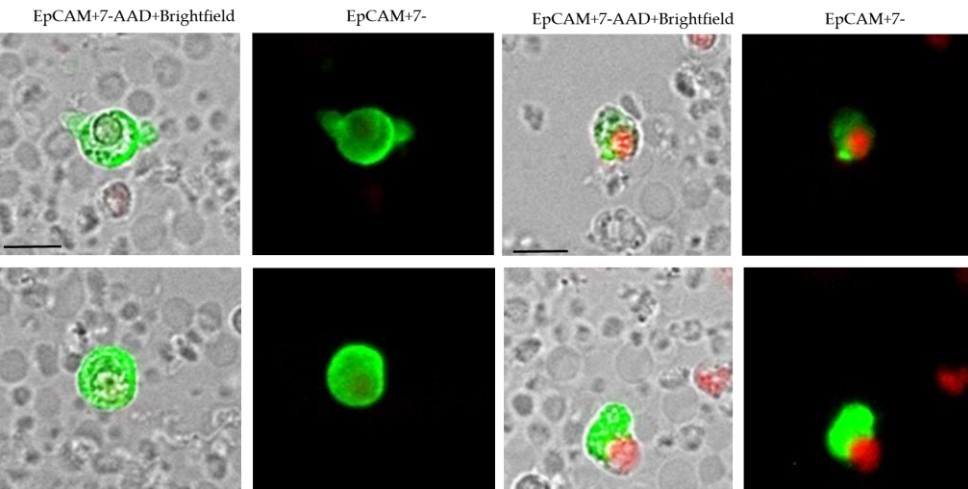

**Figure 1.** Circulating epithelial tumor cells. Viable (the four micrographs on the left) and apoptotic (the four micrographs on the right) EpCAM-positive (green) cells. In viable cells the cell membrane is impermeable preventing the nuclear dye 7-AAD (red) from entering the cell. In apoptotic cells the cell membrane becomes permeable allowing the nucleus to be stained with 7-AAD.

The study included 165 patients who received adjuvant RT either after neoadjuvant chemotherapy and BCS in the neoadjuvant setting or following adjuvant chemotherapy after BCS at Jena University Hospital between September 2002 and September 2012. Blood samples were collected a maximum of 6 weeks before and a maximum of 6 weeks after the end of RT. Patients were followed until 30 June 2022.

All patients were treated according to established guidelines and received adjuvant RT with, or without, prior systemic chemotherapy before or after BCS. Radiotherapy was given according to the guidelines as normofractionated photon radiation of the breast/thorax with a total dose of 50.0 Gy using 2.0 Gy per single dose. Patients younger than 60 years of age received a boost, up to a total dose of 66.0 Gy, normofractionated sequentially at single doses of 2.0 Gy. Patients who received radiation to the regional supra clavicular lymph nodes additionally received an additional 46.0 Gy to this anatomic location.

Statistical analysis was performed using the software program SigmaPlot version 13.0 (Systat Software Inc., Chicago, IL, USA) for Windows. Comparisons between the variables were performed with Student's *t* test (dichotomous variables) or ANOVA (variables with more than two categories), taking into account the possibility of applying nonparametric tests. Correlations were calculated with the Pearson or Spearman rank correlation coefficient. The Kaplan–Meier method was used to compare differences of relapse-free survival using a log-rank test and SigmaPlot 13. $p < 0.05$ was considered to indicate a statistically significant difference.

Ethical approval for this study was granted by the University of Jena on 13 September 2002, ethical Code 0921-08/02. The trial is registered (2 May 2019) at trials.gov under NCT03935802.

## 3. Results

For detection of cells of potential origin from solid tumors in peripheral blood, we used the surface marker EpCAM, which is expressed on epithelial cells but not on hematological cells [17–19].

We applied the same principles as those used for blood cell counting for enumeration of circulating tumor cells conducting a direct staining of viable cells with minimal loss due to omission of washing procedures [14]. Viable CETC/CTC were detected in 160 of 165 patients prior to, or after, RT (detection rate: 96.9%) (Figure 1).

CETC/CTC were prospectively counted from blood samples using the maintrac® method. Median patient follow-up was 8.97 years (range = 1.16–19.09 years). There were

31 events during this time frame, of which 12 were local recurrences and 19 were distant metastases. Patient population characteristics are shown in Table 1.

**Table 1.** Patient and disease characteristics.

| | Patient Number (%) |
|---|---|
| **Sex** | |
| Male | 0 (0.0%) |
| Female | 165 (100.0%) |
| **Age** | Mean 56.5 (min. 27–max. 89) years |
| **T stage** | |
| T1 | 98 (59.4%) |
| T2 | 56 (34.0%) |
| T3 | 6 (3.6%) |
| T4 | 5 (3.0%) |
| **N stage** | |
| N0 | 112 (67.9%) |
| N1 | 40 (24.2%) |
| N2 | 6 (3.6%) |
| N3 | 7 (4.3) |
| **Grading** | |
| 1 | 12 (7.3%) |
| 2 | 66 (40.0%) |
| 3 | 82 (49.7%) |
| Not available | 5 (3.0%) |
| **Histologic finding** | |
| IDC [1] | 122 (73.9%) |
| ILC [2]/mixed | 23 (13.9%) |
| Other | 20 (12.2%) |
| **ER [3] status** | |
| Positive | 115 (69.7%) |
| Negative | 48 (29.1%) |
| Not available | 2 (1.2%) |
| **PR [4] status** | |
| Positive | 108 (65.5%) |
| Negative | 55 (33.3) |
| Not available | 2 (1.2%) |
| **Her2/neu status** | |
| Positive | 53 (32.1%) |
| Negative | 107 (64.9%) |
| Not available | 5 (3.0%) |
| **Chemotherapy** | |
| Neoadjuvant | 44 (28.5%) |
| Adjuvant | 83 (52.1%) |
| None | 39 (19.4%) |
| **Endocrine treatment** | |
| Yes | 114 (69.1%) |
| No | 51 (30.9%) |

[1] Invasive ductal carcinoma; [2] Invasive lobular carcinoma; [3] Estrogen receptor; [4] Progesterone receptor.

Absolute counts determined before and after the course of radiotherapy were matched per patient, but no discernable pattern was seen (Supplementary Materials). Therefore, to compensate for differences in the absolute numbers of circulating tumor cells, the results

were normalized by calculating changes (increase or decrease) as the number of CETC/CTC counted after the completion of RT divided by the number of circulating tumor cells prior to the initiation of RT. Increases and decreases in CETC/CTC delineated a continuum. The diagnostic performance of decreases and increases was evaluated by constructing a receiver operating characteristic (ROC) curve and evaluated by calculating the area under the ROC curve (AUC). We found that the AUC for all patients was 0.65 ($p$ = 0.01) and for the patients with neoadjuvant treatment was 0.78 ($p$ = 0.002). At a cut-off value of 1.0, the sensitivity to predict a relapse was 58.2%, specificity 74.2% in all patients; and sensitivity 66.7%, specificity 89.7% in patients with previous neoadjuvant chemotherapy.

Using this cut-off, CETC/CTC increased in 82 patients and decreased in 83 patients over the course of RT. Of the patients with decreasing CETC/CTC, 5 suffered local recurrence and 2 developed distant metastases during the follow-up period. Among patients with increasing CETC/CTC, 7 suffered local recurrence and 16 developed distant metastases. For the statistical analysis of patient survival, only patients with a UICC stage I-IIIA tumor (8th edition of the UICC-TNM classification) were considered. A highly significant difference was found using the log-rank test based on Kaplan–Meier analysis (Figure 2) ($p$ = 0.003).

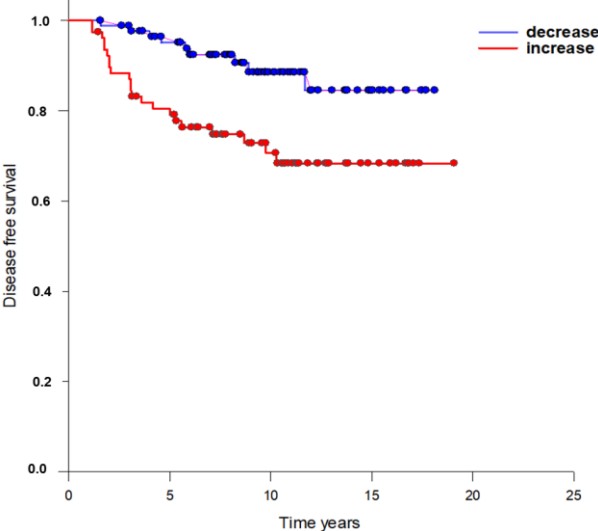

**Figure 2.** Survival curves based on Kaplan–Meier estimates for patients with early-stage breast cancer who had increased (blue) or decreased (red) numbers of CETC/CTC in the course of RT ($p$ = 0.003).

Thus, the data indicated that quantitative increases in CETC/CTC following RT was significantly associated with disease-free survival (DFS) of patients (Cox regression $p$ = 0.009; 95% confidence interval: 1.438–12.574, hazard ratio = 4.3). The risk of relapse was significantly higher in patients with increasing CETC/CTC over the course of RT when compared to patients with decreasing CETC/CTC. The five-year relapse-free survival (RFS) rate was 73.3% in patients with increasing and 90.9% in patients with decreasing CETC/CTC.

The risk factors for poor breast cancer outcomes, such as tumor size, nodal status, tumor grade, estrogen receptor status, and systemic therapy treatment, were analyzed for their relevance with DFS within this patient population. Decreases, and increases, in CETC/CTC were not differentially distributed among estrogen or progesterone receptor positive or negative tumors statistically ($p$ = 0.956). Triple-negative tumors had a tendency to recur earlier, but no further relapses occurred after three years. Further, the difference among patients with good prognostic estrogen receptor positive tumors did not reach statistical significance ($p$ = 0.063).

Larger (T2-4) tumors tended to recur earlier than T1 tumors (not shown); however, after five years, no significant differences in DFS were observed. The difference between tumor recurrence in lymph node positive and negative tumors in the study population was

also not significant. Although two relapses occurred in tumors of histological grade G1, numbers were too small to reach statistical significance, and we observed no difference between grade G2 and G3 tumors. After treatment with chemotherapy, estrogen receptor expression no longer had a positive predictive value for DFS. Hence, because of the use of risk-adapted therapy regimens, the established risk factors were no longer predictive of outcome.

Using the Cox Regression—Proportional Hazards Model neither T stage nor N-stage ER status or Her2/neu status did significantly affect DFS time.

The impact of CETC/CTC increase, or decrease, was analyzed in specific subgroups stratified by systemic treatment (Table 2).

**Table 2.** Increase and decrease in CETC/CTC numbers and relation to disease-free survival.

| CETC/CTC Behavior | Patients | Events | Local Relapse | Distant Metastasis | DFS |
|---|---|---|---|---|---|
| **No Chemo** | | | | | |
| Increase | 18 (44.4%) | 2 | 1 | 1 | $p = 0.123$ ns |
| Decrease | 21 (55.6%) | 0 | 0 | 0 | |
| **Adjuvant Chemo** | | | | | |
| Increase | 52 (54.9%) | 11 | 4 | 7 | $p = 0.172$ ns |
| Decrease | 31 (45.1%) | 3 | 1 | 2 | |
| **Neoadjuvant Chemo** | | | | | |
| Increase | 13 (38.3%) | 10 | 2 | 8 | $p < 0.001$ |
| Decrease | 31 (61.7%) | 5 | 3 | 2 | |
| **Total** | | | | | |
| Increase | 82 (47.3%) | 23 | 7 | 16 | $p = 0.003$ |
| Decrease | 83 (52.7%) | 8 | 4 | 4 | |

No chemo: no systemic chemotherapy given prior to RT; adjuvant chemo: adjuvant chemotherapy given prior to RT; neoadjuvant chemo: neoadjuvant chemotherapy given prior to RT; DFS: disease-free survival.

Thirty-nine patients enrolled in this study had surgical clearance only, without chemotherapy, due to excellent prognostic parameters. These patients had a significantly lower fraction of large tumors, significantly lower rates of lymph node involvement, less frequent high-grade (G3) histology, and less Her2/positive tumors than patients receiving chemotherapy. Of the 39 patients who had not received any chemotherapy, 21 had a decrease in CETC/CTC count and 18 had an increase in CETC/CTC count after RT as compared to before RT. In these latter patients, one local and one distant relapse occurred, but the patient number was too small to reach significance ($p = 0.123$).

Surgery followed by adjuvant chemotherapy was given to 83 patients. This group somewhat more frequently displayed T1 tumors, less frequently involved lymph nodes, a higher fraction of G2 histology, and a lower fraction of Her2/positive tumors than patients treated with neoadjuvant chemotherapy. Of the patients who were treated with adjuvant chemotherapy, 31 showed a decrease and 52 an increase in CETC/CTC counts in the course of RT. Three of the patients with a decrease in CETC/CTC counts suffered an event (3.3%), as did 11 of the patients with an increase in CETC/CTC counts (12.1%). The difference between these two groups was not statistically significant with a ($p = 0.172$) (Figure 3A).

A total of 44 patients had neoadjuvant chemotherapy followed by surgery and no other therapeutic treatment prior to RT. Of these patients 31 showed a decrease and 13 an increase in CETC/CTC counts. In the group with a decrease in CETC/CTC numbers, 5 events occurred (11.4%) during follow-up, while in the group with an increase in CETC/CTC numbers, 10 events occurred (22.7%). This was a statistically significant difference with a hazard ratio of 4.886 ($p < 0.001$) (Figure 3B).

In addition, the Kaplan–Meier analysis indicates that after less than ten years, virtually all patients treated with neoadjuvant chemotherapy whose CETC/CTC numbers increased over the course of RT were diagnosed with a relapse.

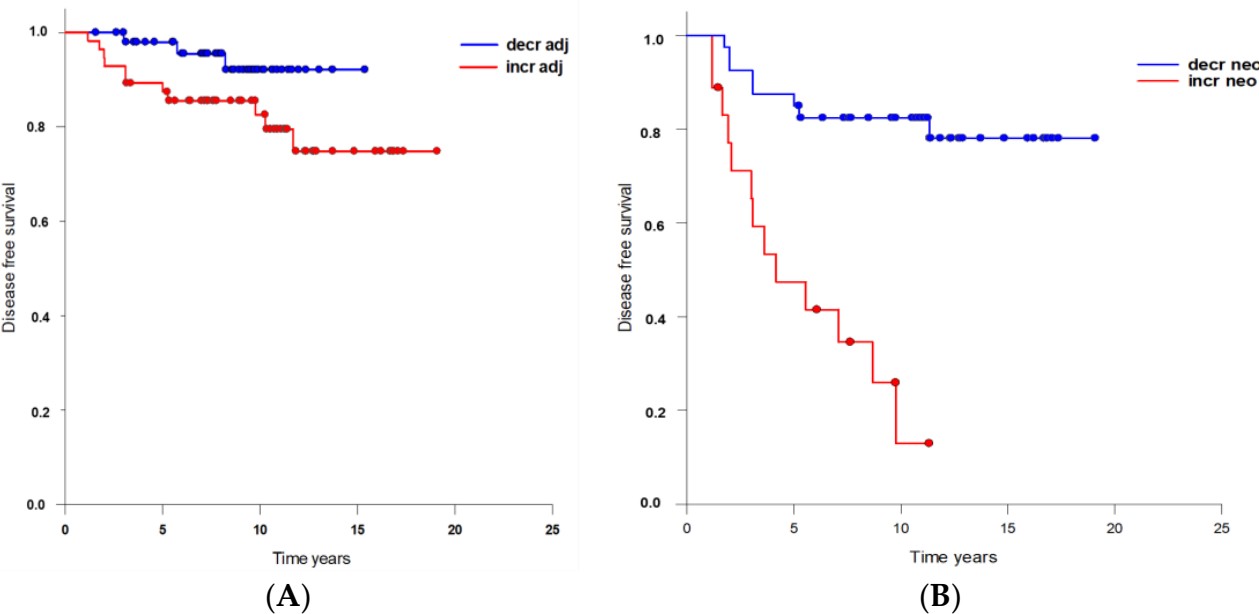

**Figure 3.** Association between (**A**) adjuvant chemotherapy (*p* = 0.172) or (**B**) neoadjuvant chemotherapy (*p* < 0.001) and DFS depending on CETC/CTC behavior over the course of RT.

In the neoadjuvant setting, there is no further therapeutic intervention between surgery and RT; therefore, the increase in cell numbers was correlated with time to relapse only in this group. The increase in cell numbers over the course of RT was inversely correlated to the time until metastases were detectable. Thus, in patients with a high increase (>10 times) of CETC/CTC numbers, metastases were detected earlier than in patients with a small increase (<10 times) (Figure 4).

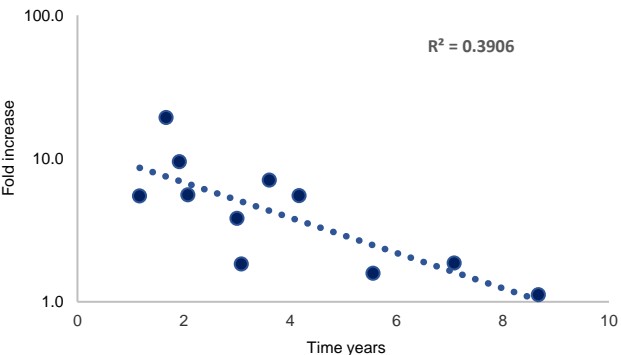

**Figure 4.** Association between the increase in CETC/CTC numbers over the course of RT and the time to metastasis detection.

## 4. Discussion

Our work highlights the relevance of CETC/CTC as predictive biomarkers, especially for patients with early-stage breast cancer and increasing cell numbers under radiotherapy.

In breast cancer, adjuvant RT to the breast is primarily intended to reduce local relapse but has been shown to reduce distant metastases in the presence, as well as in the absence, of local recurrence [20]. In addition to its local effect on microscopic residues in the tumor bed, RT also appears to have systemic effects [13,15,21].

Distant metastases are thought to develop from cells released from the primary tumor that have been able to colonize and regrow at distant sites [22]. Circulating tumor cells are assumed to be very rare. Various approaches, sometimes controversial, exist for the identification and characterization of such circulating tumor cells.

Enrichment procedures are claimed to be necessary due to the rare nature of these cells. However, even if, in such enrichment procedures, the proportion of the relevant cells is increased relative to irrelevant cells. This point has been shown in a side-by-side comparison with the CellSearch approach [23]. It is also well-established that isolation procedures can result in diminished cell yields with an up to 99% loss of the relevant events [24]. Such findings have even been observed in populations with a high fraction of the cells in question [25]. Moreover, non-epithelial blood cells are always retained during enrichment procedures, thus requiring additional identification steps for accurate evaluation. It is increasingly accepted that magnetic bead enrichment may be sub-optimal for rare cell enrichment [26]. In addition, downregulation of the EpCAM surface antigen commonly occurs during epithelial-to-mesenchymal transition [27] resulting in very low EpCAM expression and an insufficient ability to magnetically capture these cells [28]. In contrast, using an approach comparable to commonly accepted blood counting methods in which all enrichment procedures are omitted considerably more EpCAM positive cells are detected including cells with very low EpCAM expression using direct identification via immunofluorescence. This makes it possible to find higher numbers of EpCAM positive tumor cells at multiple points throughout the course of primary disease [29].

We applied this methodology to determine the number of CETC/CTC before and after adjuvant RT in an unselected patient population referred to the University Hospital of Jena during the decade from 09/2002 to 09/2012. The results of the 165 patients studied in a prospective manner were consistent with published findings [30]. All of the known prognostic markers such as tumor size, nodal involvement, triple negativity, estrogen receptor expression, and Her2/neu expression were compensated for, or even reversed by, treatment [31] after up to ten years of follow-up. During the observational period, 4 patients died as a result of the metastatic disease representing 2.5% of the total patient population included in the study. Considering only those patients who have been followed for at least 5 years following diagnosis and treatment, the relative 5-year survival rate is 94.4%.

The presence of CTC was reported in conjunction with RT in one other analysis using the CellSearch methodology [15]. In this report, relapses were observed in 23% of CTC-positive patients but 77% of relapses occurred in patients without detection of CTC. In addition, 98% of patients with detectable CTC had not suffered relapse. Another group has reported CTC release following RT, but this study did not provide clinical outcomes [12].

Our data showed a high correlation between the behavior of circulating epithelial tumor cells in the course of RT in breast cancer and DFS. An increase in CETC/CTC during adjuvant RT was predictive of both recurrence and distant metastasis. Eighty-two percent of patients who experienced an event, such as local relapse or metastasis, displayed an increase in CETC/CTC numbers in the course of RT, and only 26% had an event without an increase in CETC/CTC number. Surprisingly, the manner in which the number of CETC/CTC changed in the course of RT was the only marker that showed highly significant association with clinical outcome. Both situations, increased and decreased in CETC/CTC numbers, were observed with approximately equal frequency; however, an increase in CETC/CTC numbers was found to be a significant predictor of poor outcome (Figure 3). This was true for the subgroup of patients not treated with chemotherapy, or treated with different timing of systemic therapy. Due to a very favorable prognosis in the group not treated with chemotherapy, there were only two relapses, which occurred in the group with increasing CETC/CTC numbers, but this did not reach statistical significance. In the group of patients who had received adjuvant chemotherapy, more relapses occurred in the group with increasing numbers, but this was found to be a trend and not yet statistically significant. In the patients who had been treated with neoadjuvant chemotherapy and had the least favorable prognostic features, most of the relapses occurred with increasing CETC/CTC numbers in the course of RT, and this was statistically highly significant.

In the group receiving neoadjuvant treatment, recurrences occurred earlier than in the adjuvant chemotherapy group. The principal difference between the patient groups receiving adjuvant versus neoadjuvant chemotherapy, apart from the higher proportion of

patients with involved lymph nodes and G3 tumors, is the timing of chemotherapy, surgery, and RT. There is no therapeutic intervention between surgery and RT. Although we have shown local RT to have effects on gene expression [21], we cannot rule out that changes observed are delayed effects of systemic therapy and/or surgery and the design of our analysis does not allow us to determine whether the behavior of CETC/CTC is a direct effect of RT.

The increase in CETC/CTC must be due to the cells remaining in the body after surgery. We have shown in multiple previous studies that an increase in tumor cell numbers observed in circulation during breast disease is correlated to poor prognosis [17,32]. In patients with increasing cell numbers, a fraction of these cells could be induced into clonal growth forming, so called, tumor spheres [33]. Such spheres were recently shown to be able to form tumors on the chorion allantois membrane of chicken and thus are viewed as fully capable of forming metastases [34]. In this regard, it makes sense that a high increase in CETC/CTC numbers observed in the course of RT is associated with early recurrence. In contrast, a small increase in CETC/CTC numbers was associated with a longer latency period before recurrence became detectable. It is unclear to what extent RT itself contributes to these changes [35]. Viable cells not influenced by or resistant to RT may be mobilized into the circulation resulting in the increase in CETC/CTC observed in this study [12,36]. The correlation between the quantitative rise in CETC/CTC and the time to relapse might also signify the growth of occult tumor cells after surgery.

RT is meant to destroy local tissue at the same time activating pro-inflammatory processes [37,38]. Thus, RT may help promote the growth of distant metastases that are responsive to pro-growth signaling activated by aberrant or dysregulated changes in pro-inflammatory and pro-fibrotic cytokines [39,40]. It can take a long time for metastases to become detectable, and that this may be due to "dormant" tumor cells becoming reactivated [41,42]. However, there is also the possibility that metastases grow continuously as suggested by the increase in circulating CETC/CTC in the course of RT. We hypothesize that surgery and RT, which induce inflammatory mediators, contribute to the growth of radioresistant and aggressive tumor cells that have previously been dormant [43]. This would allow anti-inflammatory agents to reduce inflammation-induced metastasis [44]. Single cell expression profiling indicates that CETC/CTC can vary their gene expression patterns in response to treatment, and can be induced to express markers of adhesion and stem-ness [45]. Single cell expression profiling on CETC/CTC isolated before and after RT revealed that selected genes encoding proteins that function in metastatic spread, proliferation, differentiation, migration, adhesion, and apoptosis were induced in response to RT (21). A generally enhanced cell metabolism in susceptible cells in response to RT might promote metastasis. Additional work is now in progress, using predefined time points for blood withdrawal, to confirm the results in a larger patient cohort. Additionally, a controlled clinical trial is planned that will compare patients with, and without, NSAIDs during and following RT to reduce therapy-induced inflammation.

There is limited evidence that all cells identified with our method are tumor cells. So far, none of the other methods available can prove this. In addition, certainly not all CETC/CTC detected in circulation are capable of forming metastases. We previously performed single cell mutation analysis in epithelial cells isolated individually from the blood of patients with tumors with known mutations [46,47]. In this work, we demonstrated that at least some of the CETC/CTC isolated carried the same mutation as the tumor [21,48]. However, even single cell genomics cannot conclusively prove that the cells observed are truly derived from a tumor. It has recently been shown that cells from normal tissue can carry somatic mutations typical of tumor cells [49]. Further, some bona fide tumor cells can be resting cells and not relevant to the process of metastasis formation.

The results of this study may influence future treatment decisions. This is especially true in patients treated with neoadjuvant chemotherapy as an increase in CETC/CTC following RT was highly associated with tumor relapse. Follow-up using CETC/CTC as a biomarker should be considered. In addition, an interdisciplinary and coordinated

intensification of systemic therapy or RT, in terms of an individual treatment concept, is conceivable [50]. Patients with hormone receptor-positive tumors could possibly benefit from endocrine treatment lasting longer than what is currently recommended [39]. Determining CETC/CTC in the context of prospective randomized clinical trials might help to develop personalized treatment recommendations [51].

## 5. Conclusions

Using a highly sensitive method, circulating tumor cells could be detected in the peripheral blood of 96.9% of patients included in our study. A quantitative increase in CETC/CTC in the course of RT strongly correlated with an increased risk of tumor recurrence or development of distant metastasis. Based on the present results, monitoring CETC/CTC as a predictive biomarker could aid in identifying patients at increased risk of recurrence, and to better adjust therapy prior to potential disease progression. Whether the increase in CETC/CTC was due to the release of radioresistant cells from occult tumor remnants, or due to preexisting minimal residual disease remains to be investigated. In patients treated with neoadjuvant chemotherapy, which is increasingly implemented as a therapeutic approach, the increase in CETC/CTC was invariably associated with tumor recurrence, supporting the view that these patients are urgently in need of new treatment regimens. The results of this study are hypothesis-generating and highlight the need for prospective clinical trials of CETC/CTC status as a predictive criterion and to individualize treatment.

**Supplementary Materials:** The following supporting information can be downloaded at: https://www.mdpi.com/article/10.3390/curroncol30010021/s1, Figure S1: Absolute cell numbers before and after RT, sorted by quotient; Figure S2A: Course of absolute cell counts before and after RT for all 165 patients; Figure S2B: Course of absolute cell counts before and after RT of patients who received adjuvant chemotherapy; Figure S2C: Course of absolute cell counts before and after RT of patients who received neoadjuvant chemotherapy; Figure S2D: Course of absolute cell counts before and after RT of patients who did not received chemotherapy.

**Author Contributions:** Study design, conceptualization M.M. and K.P.; investigation, formal analysis, interpretation of data, writing—original draft preparation, M.M.; supervision, project administration, A.W., T.W. and S.D.; project administration, writing—review and editing, A.W. and K.P.; data curation, S.D., interpretation of data, visualization, D.S. and M.P. All authors have read and agreed to the published version of the manuscript.

**Funding:** M.M. was funded by the Deutsche Forschungsgemeinschaft (DFG, German Research Foundation) Clinician Scientist Program OrganAge (funding number 413668513) and by the Interdisciplinary Center of Clinical Research of the Medical Faculty Jena.

**Institutional Review Board Statement:** This study was conducted according to the guidelines of the Declaration of Helsinki, and approved by the Ethics Committee of the University Hospital Jena (No. 0921-08/02).

**Informed Consent Statement:** All patients provided written consent for the publication of research performed with their medical data.

**Data Availability Statement:** The datasets generated and/or analyzed during the study are not publicly available due to preservation of privacy but are available from the corresponding author on reasonable request.

**Conflicts of Interest:** K.P. is the holder of the patents for the herein described method maintrac® to detect CETCs.

## Abbreviations

| | |
|---|---|
| BCS | Breast-conserving surgery |
| CETC/CTC | Circulating epithelial tumor cells |
| DFS | Disease-free survival |
| EpCAM | Epithelial cell adhesion molecule |
| FITC | Fluoroisothiocyanate |
| GAPDH | Glyceraldehyde 3-phosphate dehydrogenase |
| RT | Radiotherapy |

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
