# Peer review of "Increased Circulating Epithelial Tumor Cells (CETC/CTC) over the Course of Adjuvant Radiotherapy Is a Predictor of Less Favorable Outcome in Patients with Early-Stage Breast Cancer"

_curroncol, doi:10.3390/curroncol30010021_

Round 1

Reviewer 1 Report

I would like to thank the authors for the chance to review their manuscript. This is a very interesting and well-written manuscript, that I really enjoyed reviewing.

The introduction is sufficient and provides adequate information. I would change the term endocrine therapy in line 42 to hormone therapy. Moreover, in line 43, the sentence should start with the number spelled and not numerical (Thirty instead of 30).

The methodology is really thorough. The only thing I would add in this section is the mention in the register of the clinicaltrials.gov (NCT03935802), which is already mentioned in the abstract. 

The results are nicely presented and do not need any alterations. However, I would like to see a footnote of the abbreviations mentioned in Table 1. Furthermore, just like in the introduction section, the sentences in lines 186 and 213, should start with the numbers spelled and not numerical. 

The discussion section is sufficient. Nonetheless, I would suggest that the authors add at the begging of the Discussion, the main findings of their study in sentence or two. Additionally, in lines 282 and 283, the authors mention Eighty-two percent, which is followed by the number (74%). This should be corrected. 

The conclusion are in concordance and supported by the results of the study.

Besides these minors flaws, this article is really nice and well presented.

Author Response

Thank you for your kind comment on our work! We have taken up your constructive suggestions for changes and revised the manuscript. We hope that you agree with this version and agree to a publication.

Reviewer 2 Report

The paper by Mäurer et al. sought to evaluate the impact of changes in circulating tumor cell (CTC) counts after radiotherapy (RT) in BC relapse and found that patients in which these counts were increased are higher risk for recurrence, especially in the context of neoadjuvant chemotherapy. Overall, the paper is of well written and holds great interest to the audience. However, some issues must be clarified prior to consider it for publication.

Major points:

1) Authors consider changes in CTC counts before and after the treatment by dividing the counts after treatment by the number found before the treatment instead of absolute numbers. However, it would be great to plot the absolute counts in both time points matched per patient to show how this data variate.

2) In the same regard, by simply dividing the counts after RT by the counts before RT, authors divide the sample into "increased" (>1) and "decreased" (<1) groups. However, this classification is not sensitive to small variations that may naturally occur due to technical reasons without representing true variations. Did the authors consider other ways to normalize the data and maybe include a "stable" group? E.g.: by calculating the delta for counts after and before and dividing this delta by the initial counts, authors would obtain proportional variations on a continuous scale that could be splitted into the suggested groups.

3) The correlation between CTC counts and clinicopathological parameters was checked? It would be interesting to check if any of these influence CTC counts.

4) Despite the authors show that no clinicopathological parameters were prognostic in the current cohort, it would be important to provide adjusted survival models considering at least those parameters that show a trend towards significance, considering that they are known to influence patients prognosis and, probably, CTC counts.

5) Is there any correlation between time for collection after RT and CTC counts? This data is important to rule out any influence that variation in time for collection could have on CTC counts.

Minor points

1) In table 1, what numbers within "[ ]" mean in the age line? Please state.

2) In the stretch "Thus, the data indicated that quantitative increases in CETC/CTC following RT was significantly associated with Disease-Free Survival (DFS) of patients (Cox regression p=0.009; 95% confidence interval: 1.438 - 12.574). The risk of relapse was also significantly higher in patients with increased CETC/CTC in the course of RT when compared to patients with decreased CETC/CTC (hazard ratio=4.3)" does the 95%CI (1.438 - 12.574) refers to the hazard ratio (4.3)? These parameters would be better reported together in the same phrase and within the same parentheses.

3) Table 2 reports the impact of CTC increase or decrease in specific subgroups stratified by treatment; however, the phrase introducing it (lines 178-180) does not makes it clear.

4) Table 3 is not cited in the text. Also, the way it is presented it is confusing. It would be better presenting treatment in table columns and compared parameters in rows.

5) In figure 5, please show p-value and correlation coefficient, making clear which correlation test was applied (r for Pearson's correlation and rho for Spearman's rank correlation).

Author Response

We thank you for your valuable comments and hope that we have comprehensively addressed all aspects in the revised version of the manuscript. We hope that the manuscript is now suitable for publication.

Major points:

1) Authors consider changes in CTC counts before and after the treatment by dividing the counts after treatment by the number found before the treatment instead of absolute numbers. However, it would be great to plot the absolute counts in both time points matched per patient to show how this data variate.

We have performed an anlysis plotting the absolute counts before and after radiotherapy matched per patient but this did not result in discernable pattern of trends. Since the analysis did not deliver any additional information it was not included into the manuscript.

2) In the same regard, by simply dividing the counts after RT by the counts before RT, authors divide the sample into "increased" (>1) and "decreased" (<1) groups. However, this classification is not sensitive to small variations that may naturally occur due to technical reasons without representing true variations. Did the authors consider other ways to normalize the data and maybe include a "stable" group? E.g.: by calculating the delta for counts after and before and dividing this delta by the initial counts, authors would obtain proportional variations on a continuous scale that could be splitted into the suggested groups.

We did perform a calculation according to the reviewers request. Values did now change from negative values (decrease) to positive values (increase). Performing the same ROC analysis on these values resulted in a cut-off of zero and using this cut-off yielded exactly the same Kaplan-Meier curves as in the initial analysis.

3) The correlation between CTC counts and clinicopathological parameters was checked? It would be interesting to check if any of these influence CTC counts.

Indeed, we checked the correlation with clinicopathological parameters. However, in this cohort no significant correlation was found. Only for the progesterone receptor was there a tendency towards a correlation (p = 0.061). In contrast, as described in the manuscript, the influence of neoadjuvant chemotherapy was significant.
Different T-stages, N-positivity, ER-positivity and Her2/neu positivity were all plotted against the cell numbers but no correlation was obtained.

4) Despite the authors show that no clinicopathological parameters were prognostic in the current cohort, it would be important to provide adjusted survival models considering at least those parameters that show a trend towards significance, considering that they are known to influence patients prognosis and, probably, CTC counts.

The Cox regression was calculated and this was included into the manuscript. Cox Regression - Proportional Hazards Model Model Estimates:

Covariate Coefficient StdErrWald Chi-SquareP Value
T_1          1,108          0,708    2,447        0,118
T_2          1,101          0,750    2,152        0,142
T_4          1,157          1,209    0,916        0,338
T_0          2,330          1,243    3,516        0,061
N_2          0,221          1,074    0,0426       0,837
N_0          0,0810         0,433    0,0350       0,852
N_3          0,394          1,099    0,129        0,720
ER_0         0,0346         0,414    0,00699      0,933
HER2_0       0,00202        0,393    0,0000265    0,996

Signficance Level = 0,050
The covariates have no significant effect on the hazard rate.

5) Is there any correlation between time for collection after RT and CTC counts? This data is important to rule out any influence that variation in time for collection could have on CTC counts.

We have also dealt intensively with this question. In 38 of the 165, the blood was taken directly on the first or last day of radiation. For the others, we defined the time period described in the manuscript. In fact, there were neither methodological differences in sample analysis (absolute cell count, cell count progression) nor differences in terms of clinical outcome. The two subpopulations did not show significant differences, so we assume that the defined time period does not significantly bias the results. In a currently ongoing prospective study, we nevertheless emphasized the importance of obtaining the samples directly on the first and last day of irradiation.

Minor points

4) Table 3 is not cited in the text. Also, the way it is presented it is confusing. It would be better presenting treatment in table columns and compared parameters in rows.

Your criticism is justified. Table 3 did not provide further information and was omitted.

5) In figure 5, please show p-value and correlation coefficient, making clear which correlation test was applied (r for Pearson's correlation and rho for Spearman's rank correlation).

In Fig.5 the trendline is shown together with the determination coefficient indicating the relationship strength between increase in cell numbers and time to relapse (Pearson's correlaion).

Round 2

Reviewer 2 Report

Authors have provided a improved version of the manuscript. However, some important issues still persist.

In the first round of review I have highlighted that it would be important to plot the absolute counts for CTCs. In my opinion it would be crucial to understand how this data varies among patients and to rule out any methodological bias in the quantification of CTCs. However, authors only state in the text that they have performed analyses considering the absolute numbers but no pattern was observed, so they have chosen to perform analysis considering the "increase" or "decrease" of CTCs, without showing any data regarding absolute counts. I think that despite no statistical significance have been observed in analyses with absolute CTC counts, showing how this data behave (e.g.: by dot plots showing CTC counts individually and data distribution before and after RT) is extremely important.

This data would also help clarify other aspects of the work, e.g.: authors state that CTCs could be detected in 160 of 165 patients either before or after RT, and that increases and decreases were calculated by dividing the CTC count after RT by the CTC count prior to treatment. Is there any patient in which no CTC was detected prior treatment? If yes, how CTC variation was calculated for these, since it would not be possible to apply the stated formula (divisor = 0)? 

Author Response

In accordance with your suggestion, we have summarised the absolute cell numbers in dot plots. At your request, we would attach the individual diagrams as a supplement to the manuscript.
In Fig. 5, the absolute cell counts before and after RT are plotted for each individual patient, sorted by the quotient. In Fig. 6A-D, we have assigned the courses individually and plotted them for the subgroups described in the manuscript. In Fig. 7A-F we have shown the cell counts for each patient individually and as a boxplot. No clear correlation or trend could be discerned over the totality of the measured values.

The detection limit in our system used for this work was 10 cells/ml. Thus, also due to the logarithmic  display 10 was chosen as the lower limit, allowing division by ten if no cells were detectable. In 5 out of 165 patients, no cells were detectable either before or after RT.

We hope that we were able to answer your questions sufficiently and that you agree to the publication of the manuscript.

Round 3

Reviewer 2 Report

Thanks to the authors for the clear responses. I think that it is important to add at least the first graph or an equivalent representation of data distribution in the main text or in the supplementary material to make the data clear to the readership.

Since all the raised concerns have been addressed, I think that the manuscript merits to be published.